# Nanotechnological Plastic Flooring: Implications for Broiler Chicken Performance, Health, and Carcass Quality

**DOI:** 10.3390/vetsci12010031

**Published:** 2025-01-08

**Authors:** Bruna Barreto Przybulinski, Rodrigo Garófallo Garcia, Maria Fernanda de Castro Burbarelli, Irenilza de Alencar Naas, Claudia Marie Komiyama, Fabiana Ribeiro Caldara, Vivian Aparecida Rios de Castilho Heiss, Kelly Mari Pires de Oliveira, Renata Pires de Araújo, Jean Kaique Valentim

**Affiliations:** 1Faculty of Agricultural Sciences, Universidade Federal da Grande Dourados, Dourados 79804-970, Brazil; brunabarreto88@gmail.com (B.B.P.); fariakita@gmail.com (M.F.d.C.B.); claudiakomiyama@ufgd.edu.br (C.M.K.); fabianacaldara@ufgd.edu.br (F.R.C.); viviancastilho@live.com (V.A.R.d.C.H.); 2Graduate Program in Production Engineering, Universidade Paulista, São Paulo 04026-002, Brazil; irenilza.naas@docente.unip.br; 3Faculty of Biological Sciences, Universidade Federal da Grande Dourados, Dourados 79804-970, Brazil; kellyoliveira@ufgd.edu.br (K.M.P.d.O.); renataaraujo@gmail.com (R.P.d.A.); 4Department of Animal Science, Universidade Federal de Viçosa, University Campus, Viçosa 36570-900, Brazil; kaique.tim@hotmail.com

**Keywords:** antimicrobial, breast callus, carcass yield, feed conversion, *Salmonella*, wood bedding

## Abstract

This study aimed to evaluate the impact of two types of plastic flooring—one with and one without nanotechnological antimicrobial additives—used as a complete or partial replacement for wood shavings on various aspects of broiler chicken production. The investigation focused on the performance, yield, meat quality, and litter microbiology in broilers raised until 42 days of age. A completely randomized design was employed using Ross 408^®^ male broiler chicks, which were distributed in five treatments that included wood shavings, plastic flooring, and combinations of both. This study included organ biometrics, the macroscopic evaluation of *Eimeria* lesions, microbiological analysis, performance indices, and meat quality parameters. The findings indicated that plastic flooring presented several challenges, such as a greater incidence of coccidiosis-related intestinal lesions, as well as negative effects on performance, body weight, and carcass yield, when compared to wood shavings.

## 1. Introduction

Broiler chickens are subjected to stress due to health issues, environmental conditions, or other factors that impact their welfare, which may compromise the structural and functional integrity of their digestive tracts [1]. In this sense, the quality of chicken litter varies microbiologically and often involves pathogenic microorganisms [2], such as *Eimeria*, which affect the gastrointestinal tracts of birds, compromising the intestinal integrity and increasing the susceptibility to secondary infections, challenging birds’ health [3]. The infection is transmitted rapidly through the oral—fecal route via the ingestion of sporulated oocysts from contaminated litter or feed [4]. Thus, litter may be considered an important source of contamination, especially for birds raised on conventional litter, which are more exposed to sanitary challenges [5]. The type of litter material plays a critical role in maintaining sanitary integrity.

Another significant pathogen in poultry farming is *Salmonella* spp., which have both health and financial implications. These bacteria may cause systemic infections, leading to poor performance or the death of animals, as well as meat contamination and food poisoning risks for consumers [6]. Infected birds excrete large amounts of *Salmonella* into the litter, which demands litter treatment to eliminate the pathogen before its reuse in poultry or agricultural settings, as chicken litter is often incorporated into the soil as waste. However, treating the litter increases the production costs and requires an extended fallowing period [7,8].

The quality of broiler meat is influenced by various factors, including genetics, sex, nutrition, rearing systems, and preslaughter management [9]. Good rearing conditions, including the choice of litter material, may enhance animal welfare and, consequently, improve the meat quality [10]. The use of vegetable-based litter materials, such as wood shavings or rice husks, has been a well-established practice in poultry farming for decades [11].

Alternative plant materials are often seasonal and require more stringent reuse protocols. Additionally, the demand for these materials is increasing as the laying hen’s production system is changing from cage-based to floor-based facilities, which require litter substrates [12].

The bedding material quality also indirectly affects broiler health and productivity [2]. Severe leg injuries caused by bacterial contamination, litter compaction, and high moisture levels may limit feed intake due to restricted mobility or pain, reducing weight gain, carcass yields, and the overall carcass quality and increasing the carcass condemnation index [13].

Considering that broiler chickens spend most of their lives on litter, developing innovative litter solutions that mitigate the growth of harmful microorganisms may be crucial for the poultry industry. It is possible that, historically, raising broiler chickens in cages or on floors has been unsuccessful due to locomotor problems and negative impacts on carcass quality [14,15]. However, advancements in manufacturing technology and the potential combination of partial plastic flooring with wood shavings may offer solutions to these challenges. Plastic flooring is already widely used in poultry production due to its washability and reusability [16,17].

Furthermore, the incorporation of nanotechnology with antimicrobial properties into plastic flooring materials may reduce surface contamination at a low cost, with proven effectiveness against bacteria such as *Salmonella enteritidis*, *Streptococcus mutans*, *Lactobacillus*, and *E. coli* [18,19]. The use of zinc oxide offers long-lasting antimicrobial action with stability and robustness [20]. Antimicrobial plastic materials are already widely used in the food and healthcare industries [21,22]. However, there is a lack of research on the application of such materials in animal production.

The possibility of using a reusable bedding material that also helps to reduce the infection pressure inside broiler sheds is, theoretically, an interesting alternative, but field tests are needed to prove whether it is possible to include this practice in the day-to-day routines of poultry farming. It is crucial to quantify plastic flooring’s impacts on carcass injuries and dermatitis to validate its use in broilers, as this alternative flooring differs from wood shavings in its interaction with birds’ legs and breasts.

This study aimed to evaluate the effects of using two types of plastic flooring (with and without nanotechnological antimicrobial properties) as a total or partial replacement for wood shavings on the performance, yield, meat quality, and microbiology of both the litter and broiler chickens raised to 42 days of age.

## 2. Materials and Methods

This study was conducted in an experimental poultry house in Dourados, Mato Grosso do Sul, Brazil, during the winter season, at a latitude of 22°13′18″ S, a longitude of 54°48′23″ W, and an altitude of 430 m. The region’s climate is classified as humid tropical with a dry winter (Cwa) [23]. The average temperature recorded was 15.5 °C (minimum 6.25 °C and maximum 25.16 °C).

### 2.1. Bird Rearing and Experimental Design

The facility was divided into pens (measuring 4 m^2^ each) with bell drinkers, tubular feeders, curtains, and overcurtains (used for initial chick warming), as well as evaporative cooling and nebulizers for internal temperature control. During the initial phase, a 250 W infrared lamp was installed in each pen. Paper was placed over the litter until the third day, after which it was removed. A chick feeder was used in each pen until the 7th day, and, from the 1st to the 10th day, a protective circle composed of wood fibers surrounded the area.

The lighting program provided 23 h of light and 1 h of darkness until the 7th day, after which the dark period was gradually extended to 6 h [24]. The lighting used 40 W lamps, providing 22 lumens/m^2^.

A total of 1500 one-day-old male broiler chicks (Ross 408^®,^, Aviagen, Brazil) with an initial average weight of 39.9 g were weighed, standardized, and randomly allocated in a completely randomized design with five treatments, namely wood shavings (WS), plastic flooring (PF), a 50/50 mix of plastic flooring and wood shavings (PF + WS), plastic flooring with nanotechnological antimicrobial additives (PFA), and a 50/50 mix of antimicrobial plastic flooring and wood shavings (PFA + WS), as shown in Figure 1. Each treatment had six replications, totaling 30 pens with 50 birds each. The birds were fed an experimental corn and soybean meal-based diet ad libitum, formulated to meet the nutritional requirements as proposed by [25]. The stocking density was 12.3 birds/m^2^, and the total area of each pen was 2.90 × 1.40 m.

### 2.2. Description of Bedding Materials

The noncommercial plastic flooring was composed of high-density polyethylene (HDPE) with UV protection. It was designed for easy assembly in the experimental pens, using interlocking plates measuring 100 × 60 cm, with 12 × 12 mm square perforations to allow excreta to pass through the plates. The floor surface was non-slip to give the birds a firm footing. The plates were suspended 20 cm above the floor, which was supported by plastic gutters. The zinc oxide antimicrobial additive was incorporated during the manufacturing process of the treated plastic flooring (MpZn_1300, TSNano, Florianópolis, Brazil), intended to have antibacterial and antifungal properties.

In the WS treatment, new heat-treated pine wood shavings (Copa Verde Maravalhas^®^, Dourados, Brazil) were distributed in each pen at a 12 cm depth. In the treatments consisting of 100% plastic flooring (PF and PFA), the total area of each pen was covered by plastic plates. In the treatments consisting of 50% wood shavings and 50% plastic flooring, the plastic flooring covered a longitudinal strip measuring 2.90 × 0.70 m, and the remaining 2.90 × 0.70 m was filled with wood shavings on either side. All wood shavings (PFA + WS, PF + WS, and WS) were turned weekly, with no material replacement, even after compaction. No floors were cleaned during the experimental period.

### 2.3. Biometrics of Organs and Eimeria Lesions

For organ biometry and *Eimeria* lesion evaluation, six birds per treatment (one per replicate) were euthanized at 7, 14, 28, and 40 days of age. Birds were selected according to the average pen weight without fasting. Birds were euthanized by cervical dislocation and were exsanguinated by cutting the jugular veins and carotid arteries. The liver, heart, spleen, and gizzard (without content) were collected for organ biometrics and weighed on a precision analytical scale (0.001 g).

*Eimeria* lesions were classified according to the *Eimeria* species and their location: *Eimeria acervulina* through lesions in the duodenum, *Eimeria maxima* through lesions between the duodenum and Meckel’s diverticulum, *Eimeria brunetti* through lesions in the ileum, and *Eimeria tenella* through lesions in the cecum. The severity of injuries was scored from 0 to 4, where (0) represented no injuries and (4) indicated severe injury, following the methodologies of Johnson and Reid [26] and Conway and McKenzie [27].

### 2.4. Microbiology and Bedding Materials

For the microbiological analysis, drag swabs were collected from the litter materials on days 0, 1, 15, 29, and 43. Sterilized probes treated with gamma rays and soaked in 1% buffered peptone water were used for collection. Disposable plastic boots were worn during the process, and shoes were placed over the boots to walk across the bedding throughout each pen. One sample was collected per replicate, resulting in a total of 30 samples. After collection, the probes were placed in sterile plastic bags, sealed, and sent for laboratory analysis.

The detection of heterotrophic microorganisms followed the ISO 4833-1:2013 method. The collected samples were weighed, placed in sterile plastic bags, and homogenized with buffered peptone water (BPW) at a 1:9 ratio (Oxoid, Hampshire, UK). Serial dilutions were then prepared up to 10^10^ in BPW, and the spread plate technique was used to inoculate the dilutions onto plate count agar (PCA). The plates were incubated at 37 °C, and, after 24 h, the colonies were counted. The results were expressed as colony-forming units per gram (CFU/g) [28] (Figure 2).

The detection of *Salmonella* spp. followed the standard method (ISO 6579:2002/Amd.1:2007). The samples were homogenized in BPW (Oxoid, Hampshire, UK) at a 1:9 ratio and incubated at 37 °C for 18–24 h for pre-enrichment. A 0.1 mL aliquot of the pre-enriched sample was transferred to 10 mL of Rappaport–Vassiliadis (RV) broth (Merck KGaA, Darmstadt, Germany) and incubated at 42 °C for 24 h. Additionally, a 1 mL aliquot was inoculated into 10 mL of tetrathionate (TT) broth (Merck KGaA, Darmstadt, Germany) and incubated at 37 °C for 24 h.

After incubation, a loop of both the RV and TT enrichment broth cultures was streaked onto plates containing Hektoen agar (Merck KGaA, Darmstadt, Germany) and incubated at 37 °C for 24 h. Colonies displaying a black center with a transparent halo were isolated for later biochemical tests for confirmation [28] (Figure 3).

### 2.5. Performance and Carcass and Cut Yields

Birds and feed waste were weighed weekly for feed consumption, weight gain, and feed conversion evaluations. The viability of each pen was also evaluated weekly. The performance periods presented were 1 to 7 days, 1 to 21 days, and 1 to 42 days.

At 42 days of age, 108 birds, three birds per replicate, selected by the average pen weight (±10%) after a 6 h fasting period, were euthanized by cervical dislocation and were exsanguinated by cutting the jugular veins and carotid arteries. Bleeding was allowed for three minutes, followed by scalding at 58 °C, plucking, evisceration, and removal of the feet and head.

After evisceration, the hot carcass weight (HCW) was recorded. All carcasses were subsequently subjected to prechilling for 18 min (10 to 18 °C), followed by chilling for 12 min (0 to 2 °C) to cool the carcasses. The carcasses were then weighed to obtain the chilled carcass weight (CCW). The hot carcass yield (HCY) was calculated via Equation (1), and the yields of various cuts—including the breast, leg, wing, back, and boneless legs—were measured based on the CCW.
HCY = HCW × 100/BW(1)

### 2.6. Injuries, Incidence of Myopathy, and Meat Quality

After the carcasses were cooled, the incidence of injuries, including contact dermatitis, dorsal scratches (recent and old), chest calluses, and bruises, was assessed. Injuries were evaluated via a scoring system, consisting of 0 (no injury), 1 (moderate injury), and 2 (severe injury), according to each category, following the methodology proposed by Martrenchar et al. [29].

After cutting and deboning, the breast fillets were maintained for 24 h at 5 °C for the evaluation of the incidence of wooden breast and white striping myopathies, as well as the fillet dimensions, pH, colorimetry, water retention capacity, and drip loss.

Myopathies were assessed macroscopically, visually, and by palpation. Wooden breast and wooden thigh myopathies were graded as follows: 0 = absence of myopathy, 1 = intermediate or moderate, and 2 = severe [30]. For white striping myopathy, the scale was as follows: 0 = normal (no white lines), 1 = moderate (thin white lines, less than 1 mm), 2 = severe (thick white lines, 1 to 2 mm), and 3 = extreme (very thick white lines, over 2 mm, covering most of the fillet surface) [31]. The incidence of wooden thigh and white striping myopathies in the boneless thigh muscle was also evaluated via the same methodology as applied to the breast fillets.

The width (cm), length (cm), and thickness (mm) of the breast fillets were measured via a digital caliper [32]. The breast fillet pH was measured using a Testo 205 digital pH meter calibrated with buffer solutions (pH 4.0 and 7.0, ±0.05 at 25 °C). The measurements were taken at three different points on the breast, and the average value was recorded [33].

For the meat color analysis, the fillets were kept at room temperature for 30 min and then a Minolta CR 400 portable colorimeter was used in the CIELab system to measure the L* (lightness), a* (redness), and b* (yellowness) parameters, using the average of three points per sample [33].

To evaluate the water retention capacity (WRC), 2 g of each sample was weighed, placed between two filter papers and glass plates, and pressed for 5 min under a 10 kg weight. The sample was reweighed after pressing, and the difference in the weights was used to calculate the percentage of water retention [34].

Drip loss (DL) was assessed by placing 80 g samples in a mesh suspended inside a plastic container at 4 °C for 48 h and then reweighing the samples. The percentage loss due to exudation was determined by the difference between the final and initial weights of the sample [35].

The thawing loss, cooking loss (CL), and shear force were analyzed after the samples were frozen at −20 °C for 30 days. Thawing loss was calculated as the difference between the frozen weight and the weight after the sample was refrigerated at 4 °C for 24 h [36]. For cooking loss, the breast fillet samples were weighed, sealed in plastic bags, and cooked in a water bath at 85 °C for 30 min. After cooking, the fillets were cooled to room temperature, and the difference between the fresh and cooked weights was recorded as the cooking loss [36].

The shear force was measured via an AXT 2i texture analyzer (Stable Micro Systems). After the cooking analysis, five samples per breast, each measuring 1 × 1 × 2 cm, were cut and placed between the blades of the texture analyzer, with the fibers positioned perpendicularly. The blade descended at a speed of 200 mm/min. The shear force for each breast was determined by averaging the five measurements [37].

### 2.7. Statistical Analysis

Organ biometric data, heterotrophic microorganisms, performance, meat quality, and carcass yields were evaluated for normality of residuals via the Shapiro–Wilk test and homogeneity of variances via Levene’s test. Thus, an analysis of variance was performed via PROC MIXED from [38]. The statistical models included the type of floor as a fixed factor and the animal effect, nested within each treatment, included as a random factor in all models (RANDOM command). When significant, the Tukey test was performed to compare the treatment means, considering the 5% level.

For the statistical analysis of the presence of *Salmonella* in the bedding materials, as it was a categorical variable, the chi-squared test or Fisher’s exact test was performed via the SAS FREQ procedure [38], with a significance level of 5%. The data in the tables are expressed as a percentage of the samples evaluated from each treatment, followed by the integer corresponding to this percentage.

A descriptive data analysis was carried out during the observations of the *Eimeria* lesions. The data obtained for the assessments of myopathies did not have a normal distribution. Thus, the theory of generalized linear models proposed by Nelder and Wenderburn [39] was used, and the SAS GLIMMIX procedure was applied [38]. The data distribution was GAMMA, except for data on spondylolisthesis and digital fractures, in which the BINARY distribution was used. For data on carcass injuries, as they were presence or absence data, a binary distribution was adopted. The GAMMA distribution assumes that the data residuals have exponential behavior, whereas the BINARY distribution assumes that the residuals have Poisson behavior.

## 3. Results

No significant differences in broiler performance were detected between the treatments up to 7 days of age (Table 1). Plastic flooring did not affect feed consumption or bird viability up to 21 days of age. However, wood shavings played a crucial role in weight gain. Compared with those raised exclusively on plastic flooring (PF or PFA), broilers raised on wood shaving litter (WS, PF + WS, or PFA + WS) exhibited greater weight gain at 21 days (*p* = 0.002). At this stage, birds raised on PF demonstrated poorer feed conversion than those raised on PFA + WS, although no significant differences were observed with the other treatments (*p* = 0.0114).

By 42 days of age, birds in the PF treatment consumed less feed than those in the PF + WS and PFA + WS groups did (*p* = 0.0012). In addition to the reduced feed intake, birds raised on plastic flooring (PF and PFA) also presented poorer weight gain and feed conversion (*p* < 0.0001), along with lower viability (*p* < 0.0001). These findings underscore the importance of the substrate choice, with wood shaving litter being a critical component for optimal broiler performance until 42 days of age.

At 14 days, birds raised on PFA presented greater liver weights than those raised on WS or PFA + WS (Table 2). Additionally, birds reared on wood shavings had the lowest gizzard weights. After 28 days, the bedding materials had no significant effect on the organ weight. However, at 40 days, birds raised on WS presented greater gizzard weights than those raised on PF, with no significant differences observed in the weights of the other organs.

At seven days of age, no lesions characteristic of *Eimeria* spp. were observed in the broilers’ intestines. By 14 days, one bird in the PF + WS treatment group presented lesions indicative of *Eimeria brunetti* (grade 2), with numerous petechiae on the serous surface, primarily in the ileum, and fewer petechiae on the mucosal surface. At 28 days, two birds from the PFA treatment group and one bird each from the PF, PF + WS, and PFA + WS treatment groups presented lesions characteristic of *Eimeria brunetti* (grade 1). By 42 days, lesions consistent with *Eimeria brunetti* (grade 1) were identified in two birds from the WS treatment, one from PFA + WS, and one from PF + WS.

Additionally, four birds from the PF treatment group presented lesions indicative of *Eimeria maxima* (grade 3), marked by the thickening of the intestinal wall, abundant orange mucus, and traces of blood. One bird from the WS treatment group and one from the PFA + WS treatment group presented lesions characteristic of *Eimeria maxima* (grade 2), including thickened intestinal walls, petechiae on the serosal surface, smaller petechiae on the mucosal surface, and orange mucus in the intestinal content (Figure 4).

Analyzing the count of heterotrophic bacteria, no significant differences were observed between the treatments, with all bedding materials exhibiting similar bacterial behavior over time (Figure 5). The presence of birds on the litter was a key factor in the increase in the number of mesophiles during the first 15 days. Additionally, the antimicrobial additive in the plastic flooring proved ineffective in reducing the microbial population in this study.

The bedding material significantly affected the presence of *Salmonella* at 15 days (*p* = 0.0227), with *Salmonella* detected in 33.33% of the PF samples and 16.67% of the PFA and PFA + WS samples, whereas no *Salmonella* was detected in the other treatments. During the other evaluation periods, the incidence of bacteria was low, with no *Salmonella* detected in the collected samples (Table 3).

The body weights, hot carcass weights, and cooled carcass weights of birds slaughtered at 42 days differed significantly (*p* < 0.0001), with birds raised on plastic flooring (PF and PFA) having lower weights (Table 4). No significant effect of the treatments was observed on the cut yields, except for the hot carcass yield (*p* = 0.0019) and chilled carcass yield (*p* = 0.0108), where birds reared on wood shavings (WS) achieved higher yields than those raised on PF and PFA (Table 5).

The litter material did not cause injuries such as breast callus, scratches (old or recent), hematoma, or carcass dermatitis (Table 6). The flooring material did not influence thigh injuries (white striping and wooden thighs) or white striping on the breast fillet. However, in the case of wooden breast lesions, birds raised on PFA had a lower incidence than those raised on WS or PFA + WS (*p* = 0.0023) (Figure 6).

In terms of meat quality, no significant differences were observed for the variables L* (lightness), the water retention capacity, the exudation loss, the weight loss due to thawing, the weight loss due to cooking, or the shear force (Table 7). However, the smallest breast fillet dimensions were found in birds raised on PF and PFA (length *p* = 0.0282; width *p* < 0.0001; thickness *p* = 0.0007). Compared with those from the birds in the PFA treatment, those from the WS and PFA + WS treatments presented higher pH and yellow content values (*p* = 0.0051 and *p* = 0.0096, respectively). The red content of the fillets from the WS and PFA + WS treatments was lower than that of those from the PF treatment (*p* = 0.0096).

## 4. Discussion

The influence of the litter material on broiler chicken health, performance, and organ development is multifaceted, with the environment playing a critical role in determining feed intake, mobility, and overall growth. In particular, the weight of the gizzard, which is closely tied to muscular activity and food consumption, serves as an important indicator of how different litter materials affect the broiler physiology. In this study, birds raised on wood shavings presented greater gizzard weights at 40 days, likely due to better mobility and access to feed than those raised on plastic flooring.

Another possibility is related to the natural behavior of birds in search of structural material in the environment for ingestion and to aid in the development of gizzard functions. Birds have the habit of consuming the bedding material, which influences mechanical digestion and improves the digestibility of the digesta [40]. The environment with plastic flooring reduced the availability of structural material, which may have reduced the digestive capacity of the birds and consequently their performance.

Additionally, the effects of the litter materials extend beyond organ development to microbial presence, with *Salmonella* and *Eimeria* species posing significant sanitary challenges depending on the type of litter used. These results show the complexity of selecting appropriate bedding materials, as they impact not only chicken growth and health but also the microbiological quality of the rearing environment.

The weight of the gizzard is closely linked to its muscular activity and the amount of feed consumed, with the birds’ environment being a key factor influencing feed intake and, consequently, the gizzard weight. At 40 days, birds raised on wood shavings presented greater gizzard weights, which correlated with their greater body weights.

In contrast, birds raised on plastic flooring were less mobile in terms of access to feed and spent more time sitting. Similarly, Brito et al. [41] reported no differences in the viscera weight between treatments when broiler chickens were evaluated on different types of litter, attributing the lower weight gain to reduced movement and feed consumption when the birds rested on specific bedding materials.

No *Eimeria* lesions were observed at seven days, with only one bird affected at 14 days, likely because the peak in *Eimeria* infestation occurred after four weeks of housing [42]. Younger birds are more susceptible to clinical infections because of their immature immune systems, whereas older birds are asymptomatic, allowing for the spread of the protozoan and increased outbreaks [43].

In this study, only a few lesions were characteristic of *Eimeria brunetti*. In commercial broiler production, seven species of *Eimeria* are recognized as infective, each of which varies in pathogenicity. *E. necatrix* and *E. tenella* are considered the most pathogenic, causing intestinal hemorrhage with high morbidity and mortality rates. Moreover, *E. acervulina*, *E. brunetti*, and *E. maxima* cause clinical disease, and *E. mitis* and *E. praecox* are nonpathogenic but may lead to increased feed conversion and reduced growth [44]. Gazoni et al. [42] reported a greater incidence of *E. acervulina* lesions, followed by *E. maxima*, on Brazilian broiler farms, which were positively correlated with cell desquamation, litter ingestion, and feed passage.

The lack of differentiation in heterotrophic bacteria between the litter materials may be attributed to the low contamination levels in the experimental environment, reducing the potential for sanitary challenges that could influence microbial behavior. Additionally, the low environmental temperature probably reduced bacterial proliferation. In this sense, Soliman and Hassan [45] concluded that plastic flooring and cages may help to maintain the air quality and reduce bacterial contamination.

Similarly, Heitmann et al. [46] reported no significant effect of partial plastic flooring on bacteria (coliforms, *E. coli*, and ESBL-producing bacteria) in broiler production. *Salmonella* was detected in 70.1% of floor samples and 74.4% of wood bedding samples from breeder aviaries, indicating that there was no microbiological difference between the materials [47].

In this study, the nanotechnological antimicrobial flooring did not exhibit significant antimicrobial effects. One option to improve its efficiency would be to test the long-term reuse of the floor and its ability to inhibit biofilm formation. The effectiveness of antimicrobials depends on several factors. The action of the zinc oxide-based nanomaterial used here involves the migration of metal ions that compromise the microbial membrane integrity [18].

Zinc oxide nanoparticles demonstrate strong antimicrobial action against both Gram-positive and Gram-negative bacteria, although they are more effective against Gram-positive bacteria [48]. *Salmonella*, which is Gram-negative, is relatively resistant to zinc oxide nanoparticles. Another hypothesis is that biofilms may represent an additional form of resistance to bacteria [48], because they are able to form protection through their compounds, causing antimicrobial compounds to be inactivated even before establishing contact with the microorganisms.

The wood shaving litter played a critical role in broiler performance, particularly after 21 days of age. Compared with those raised on plastic flooring (PF or PFA), those raised on wood shavings presented better feed conversion and viability by the end of the experimental period. In contrast to these findings, previous studies have reported no differences in broiler performance between conventional litter and plastic flooring [49,50,51,52].

The poorer feed consumption, weight gain, feed conversion, and viability observed in birds raised on plastic flooring may be due to reduced movement and increased sitting time, leading to decreased performance. Birds raised on wood shavings tend to be more active, as this material provides an environment more akin to their natural conditions [53].

The low environmental temperature during the study (minimum of 6.25 °C, maximum of 25.16 °C) likely contributed to the higher mortality, particularly in birds raised on plastic flooring (PF and PFA). Plastic flooring distributes heat less uniformly than wood shavings [49], and extreme cold is a known risk factor for pulmonary hypertension syndrome (ascites) [54]. Compared with those raised on wood shavings, those raised on plastic flooring (PF or PFA) were heavier, but they presented lower carcass yields and smaller breast fillet dimensions. However, the bedding materials did not significantly affect the cut yield. Antimicrobial flooring did not improve the performance or carcass yield.

Based on these findings, it was possible to verify that the plastic floor yielded satisfactory results for the birds up to 21 days of age; however, as the birds become older and heavier, the plastic floor appears to compromise their performance, culminating in lower flock viability, which seems to have been aggravated by the thermal challenge faced by the chicks during the initial few days.

Contrary to our expectations, there was no significant difference in carcass lesions between the litter materials. Previous studies have suggested that the litter influences breast callus lesions [49,54], but this was not observed here. The lower environmental temperature may have mitigated this issue, as breast calluses were more common among birds raised on plastic floors during the summer [50].

Wooden breast lesions are multifactorial, with increased incidence linked to greater weight gain and pectoral muscle development [55,56]. Compared with birds raised on plastic flooring, those raised on WS and PFA + WS had a greater incidence of wooden breast myopathy, corresponding with greater body weights, carcass yields, and breast dimensions.

Broilers raised on PF had higher red meat content than those raised on WS and PFA + WS did, which differs from the results of Özbek et al. [9], who reported higher red content in chickens raised on rice husks than in those raised in open-air systems and on plastic flooring. The meat of the birds raised on WS and PFA + WS was relatively yellow. Meat color is an important factor for consumers, influencing product choices. While red meat is generally preferred, a high b* value (yellow content) is less desirable [57], making PF treatment beneficial for this variable.

Although the pH values were within the ideal range for chicken meat (5.6–6.4) [58], the higher pH observed in the breasts of birds raised on WS and PFA + WS may be related to the breast size, as there is a known correlation between larger pectoral muscles and a higher pH due to lower glycolytic potential [52,59,60]. However, other studies have reported a higher pH in the meat of chickens raised on plastic flooring compared to wood shavings [9,52]. Additionally, the use of plastic flooring as chicken litter demonstrated challenges, such as a higher incidence of intestinal lesions characteristic of coccidiosis, and the antimicrobial additive did not have the expected effect in reducing pathogenic microorganisms.

When used as a total replacement for wood shavings, plastic flooring demonstrated limitations, negatively impacting the performance, body weight, and carcass yields of broilers at up to 42 days of age. A potential alternative could be the combined use of plastic flooring with wood shavings, which maintains the performance, carcass yield, and meat quality while mitigating the adverse effects observed with the exclusive use of plastic flooring.

## 5. Conclusions

The total replacement of wood shavings with plastic flooring is not recommended due to reduced performance and viability. Plastic flooring for broiler chickens presented some challenges, such a higher incidence of coccidiosis and no significant reduction in pathogens with the antimicrobial additive, even lacking an impact on the cut yields or carcass injuries. Combining plastic flooring with wood shavings may offer a balanced solution, maintaining the performance and meat quality. It is important to note that the flooring used in this research was not a commercial product but rather flooring that is undergoing adaptation to obtain a product that meets the needs of poultry. The ideal type of flooring has not yet been found, but these results are important for the continued development of the ideal product.

## Figures and Tables

**Figure 1 vetsci-12-00031-f001:**
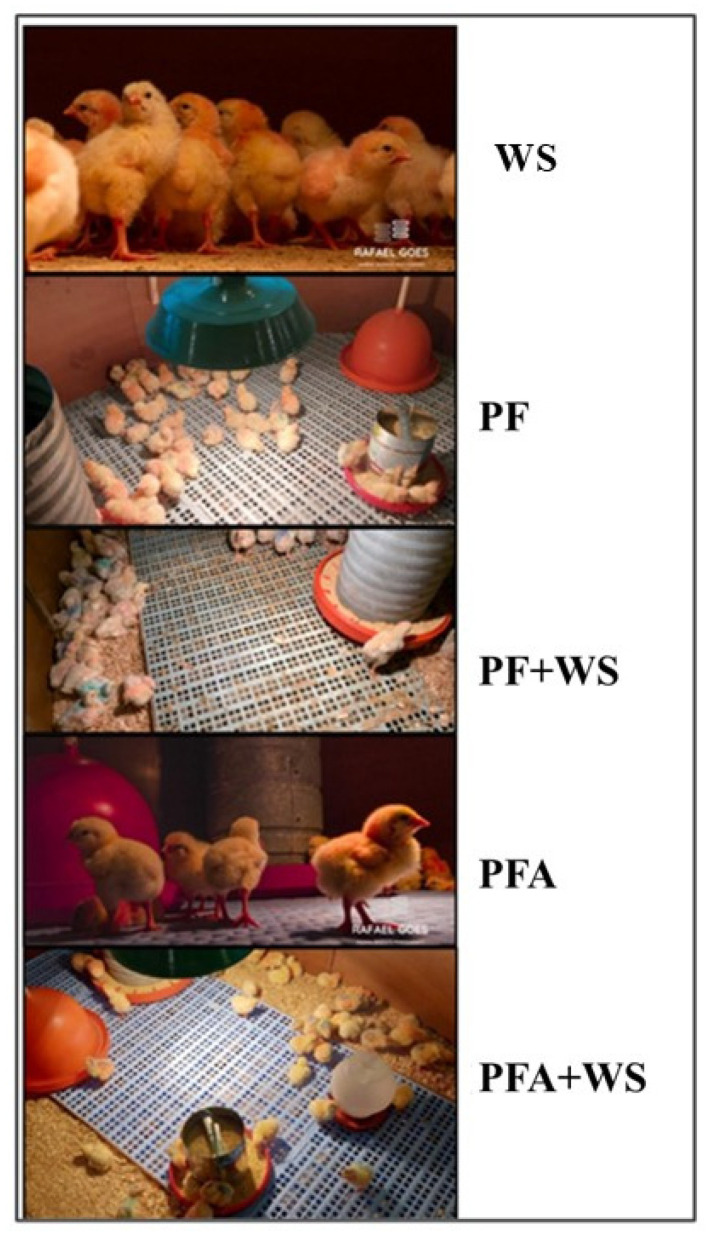
The treatments used in the experiment were as follows: wood shavings (WS); plastic flooring (PF); 50% plastic flooring and 50% wood shavings (PF + WS); plastic flooring with a nanotechnological antimicrobial additive (PFA); and 50% plastic flooring with a nanotechnological antimicrobial additive and 50% wood shavings (PFA + WS).

**Figure 2 vetsci-12-00031-f002:**
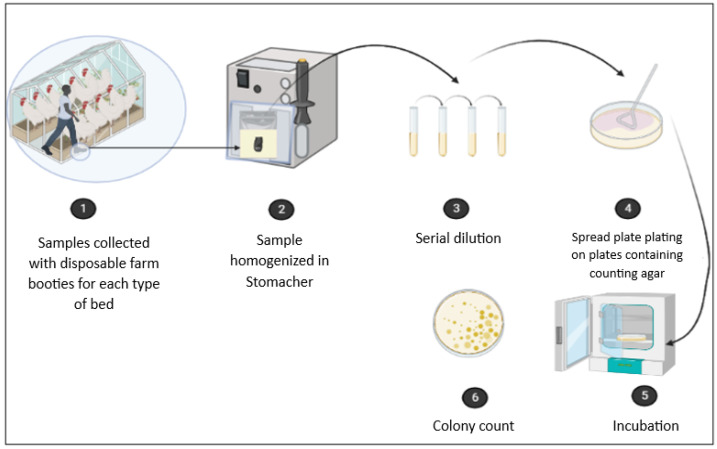
Schematic of the heterotrophic research methodology.

**Figure 3 vetsci-12-00031-f003:**
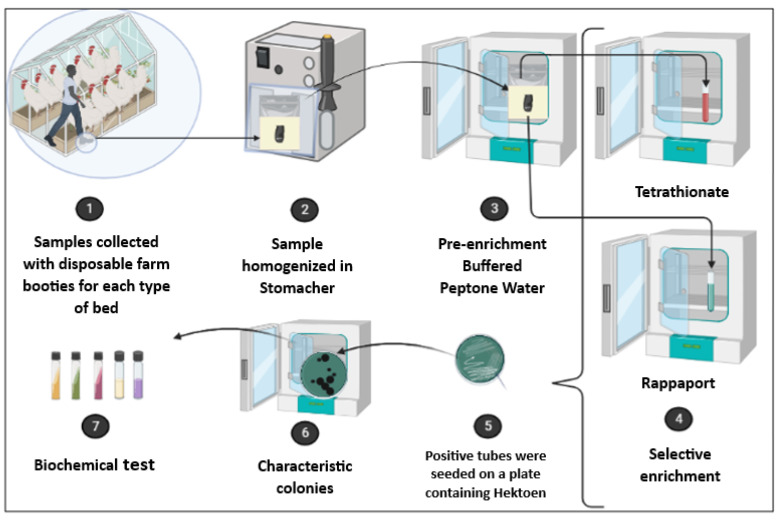
Schematic of the *Salmonella* research methodology.

**Figure 4 vetsci-12-00031-f004:**
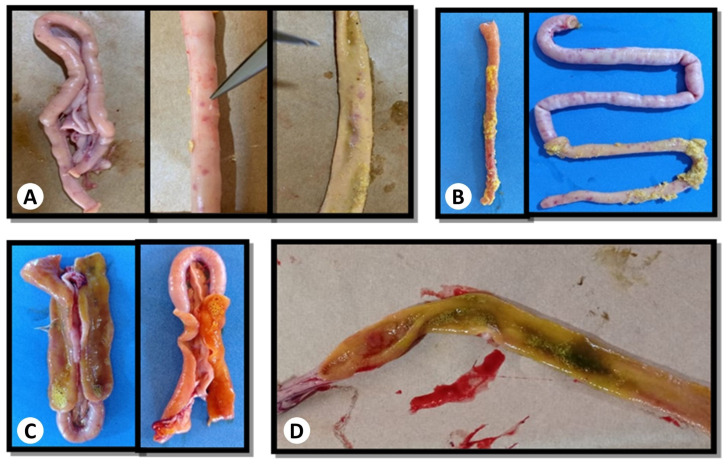
Macroscopic lesions of *Eimeria* in broiler chickens: (**A**) lesions characteristic of *Eimeria brunetti* grade 1; (**B**) lesions characteristic of *Eimeria brunetti* grade 2; (**C**) lesions characteristic of *Eimeria maxima* grade 2; (**D**) lesions characteristic of *Eimeria maxima* grade 3. Source: private file.

**Figure 5 vetsci-12-00031-f005:**
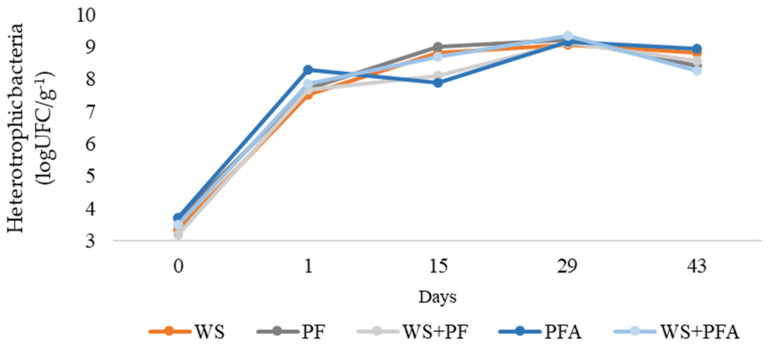
Counts of heterotrophic bacteria (log CFU/g^−1^) in different broiler litter materials: wood shavings (WS), plastic flooring (PF), 50% wood shavings and 50% plastic flooring (PF + WS), plastic flooring with nanotechnological antimicrobial additive (PFA), and 50% wood shavings and 50% plastic flooring with nanotechnological antimicrobial additive (PF + WSA). Equations for the breakdown of treatment × time interactions and standard errors of determination coefficients: WS: y = 0.00005739x^3^ − 0.00832x^2^ + 0.38407x + 3.38172 (*p* = 0.0099); PF: y = 0.000043x^3^ − 0.00728x^2^ + 0.36137x + 372631 (*p* = 0.0156); PF + WS: y = −0.00000574x^4^ + 0.00074996x^3^ − 0.03384x^2^ + 0.65923x + 3.21833 (*p* = 0.0002); PFA: y = −0.00000825x^4^ + 0.00107x^3^ − 0.04615x^2^ + 0.78322x + 3.76 (*p* = 0.0002); PFA + WS: y = −0.00000384x4 + 0.00050523x^3^ − 0.02449x^2^ + 0.56249x + 3.37667 (*p* = 0.0106).

**Figure 6 vetsci-12-00031-f006:**
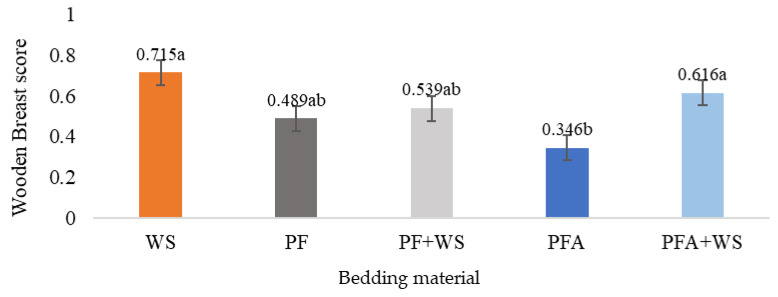
Wooden breast injury scores of 42-day-old broiler chickens raised on different bedding materials. Effect of floor type on the wooden breast score (least squares mean ± the retransformed standard error; higher values indicate greater abnormalities). Different letters (a, b) indicate statistically significant differences at *p* < 0.05. WS: wood shavings; PF: plastic flooring; PF + WS: 50% wood shavings and 50% plastic flooring; PFA: plastic flooring with nanotechnological antimicrobial additive; PFA + WS: 50% wood shavings and 50% plastic flooring with nanotechnological antimicrobial additive.

**Table 1 vetsci-12-00031-t001:** Performance of broiler chickens raised on different bedding materials.

Variable	Bedding Material					SEM	*p* Value
	WS	PF	PF + WS	PFA	PFA + WS		
	1 to 7 Days						
Feed consumption (g)	141.56	146.38	146.64	146.86	149.29	1.497	0.6147
Weight gain (g)	127.13	121.42	127.94	124.15	130.41	1.219	0.1561
Feed conversion (g/g)	1.113	1.206	1.148	1.181	1.148	0.014	0.3294
Viability (%)	98.00	98.00	98.67	97.67	100.00	0.341	0.2025
			1 to 21 days				
Feed consumption (g)	1221.08	1269.62	1268.30	1287.03	1270.07	0.282	0.5422
Weight gain (g)	863.95 a	791.63 c	858.47 ab	804.53 bc	883.88 a	1.62	0.0002
Feed conversion (g/g)	1.471 ab	1.607 a	1.478 ab	1.599 ab	1.437 b	0.751	0.0114
Viability (%)	93.35	90.67	93.66	91.95	95.66	0.745	0.2837
	1 to 42 days						
Feed consumption (g)	4537.92 abc	4201.42 c	4773.83 a	4354.18 bc	4638.38 ab	1.432	0.0012
Weight gain (g)	2709.15 a	2009.52 b	2709.50 a	2099.85 b	2766.73 a	0.307	<0.0001
Feed conversion (g/g)	1.679 b	2.098 a	1.765 b	2.081 a	1.677 b	0.060	<0.0001
Viability (%)	76.75 a	55.33 b	79.00 a	55.83 b	79.66 a	1.804	<0.0001

Average values in the same row followed by different lowercase letters indicate significant differences (*p* < 0.05). WS: wood shavings; PF: plastic flooring; PF + WS: 50% plastic flooring and 50% wood shavings; PFA: plastic flooring with nanotechnological antimicrobial additive; PFA + WS: 50% plastic flooring with nanotechnological antimicrobial additive and 50% wood shavings. SEM: standard error of the mean.

**Table 2 vetsci-12-00031-t002:** Biometrics of the organs (g) of broiler chickens raised on different bedding materials.

Organ	Bedding Material					SEM	*p* Value
	WS	PF	PF + WS	PFA	PFA + WS		
	14 days						
Heart	3.39	3.99	3.73	3.89	4.14	0.098	0.1372
Liver	14.43 c	18.01 ab	18.50 ab	18.71 a	16.11 bc	0.407	0.0002
Spleen	0.48	0.51	0.38	0.44	0.57	0.031	0.4686
Gizzard	10.57 b	11.85 ab	12.792 a	13.19 a	13.45 a	0.260	0.0005
	28 days						
Hear	10.15	10.91	10.47	11.32	10.64	0.305	0.8163
Liver	37.35	43.04	40.10	41.88	40.16	0.808	0.2261
Spleen	1.72	1.76	1.64	1.58	1.82	0.076	0.8955
Gizzard	30.62	27.94	30.65	28.16	31.23	0.672	0.3948
	40 days						
Heart	16.42	17.22	17.01	17.31	16.63	0.286	0.7941
Liver	59.08	64.32	60.00	62.07	58.90	1.162	0.2482
Spleen	3.08	3.03	3.34	2.85	3.16	0.106	0.6964
Gizzard	40.51 a	35.17 b	38.73 ab	35.70 ab	39.71 ab	0.673	0.0217

Average values in the same row followed by different lowercase letters indicate significant differences (*p* < 0.05). WS: wood shavings; PF: plastic flooring; PF + WS: 50% plastic flooring and 50% wood shavings; PFA: plastic flooring with nanotechnological antimicrobial additive; PFA + WS: 50% plastic flooring with nanotechnological antimicrobial additive and 50% wood shavings. SEM: standard error of the mean.

**Table 3 vetsci-12-00031-t003:** Presence of *Salmonella* spp. (%) in different broiler litter materials.

Day	Result	Bedding Material					*p* Value
		WS	PF	PF + WS	PFA	PFA + WS	
0	Absent	100 (6/6)	83.33 (5/6)	100 (6/6)	83.33 (5/6)	100 (6/6)	0.0828
	Present	0 (0/6)	16.67 (1/5)	0 (0/6)	16.67 (1/5)	0 (0/6)	
1	Absent	100 (6/6)	100 (6/6)	100 (6/6)	100 (6/6)	100 (6/6)	1.0000
	Present	0 (0/6)	0 (0/6)	0 (0/6)	0 (0/6)	0 (0/6)	
15	Absent	100 (6/6)	66.67 (4/6)	100 (6/6)	83.33 (5/6)	83.33 (5/6)	0.0227
	Present	0 (0/6)	33.33 (2/6)	0 (0/6)	16.67 (1/5)	16.67 (1/5)	
29	Absent	100 (6/6)	100 (6/6)	100 (6/6)	100 (6/6)	100 (6/6)	1.0000
	Present	0 (0/6)	0 (0/6)	0 (0/6)	0 (0/6)	0 (0/6)	
43	Absent	100 (6/6)	100 (6/6)	100 (6/6)	100 (6/6)	100 (6/6)	1.0000
	Present	0 (0/6)	0 (0/6)	0 (0/6)	0 (0/6)	0 (0/6)	

Results are expressed as a percentage of the samples evaluated from each treatment, followed by the integer corresponding to this percentage. WS: wood shavings; PF: plastic flooring; PF + WS: 50% wood shavings and 50% plastic flooring; PFA: plastic flooring with nanotechnological antimicrobial additive; PFA + WS: 50% wood shavings and 50% plastic flooring with nanotechnological antimicrobial additive.

**Table 4 vetsci-12-00031-t004:** Body weights (BW), hot carcass weights (HCW), and chilled carcass weights (CCW) of 42-day-old broiler chickens raised on different bedding materials.

Variable	Bedding Material					SEM	*p* Value
	WS	PF	PF + WS	PFA	PFA + WS		
BW (kg)	3.17 ab	2.72 c	3.07 b	2.73 c	3.21 a	0.026	<0.0001
HCW (kg)	2.44 a	2.06 b	2.34 a	2.07 b	2.42 a	0.023	<0.0001
CCW (kg)	2.44 ab	2.05 c	2.32 b	2.05 c	2.46 a	0.023	<0.0001

Average values in the same row followed by different lowercase letters differ significantly (*p* < 0.05). WS: wood shavings; PF: plastic flooring; PF + WS: 50% wood shavings and 50% plastic flooring; PFA: plastic flooring with nanotechnological antimicrobial additive; PFA + WS: 50% wood shavings and 50% plastic flooring with nanotechnological antimicrobial additive. SEM: standard error of the mean.

**Table 5 vetsci-12-00031-t005:** Cut yields (CY), hot carcass yields (HCY), and chilled carcass yields (CCY) of broiler chickens raised on different bedding materials at 42 days.

Variable	Bedding Material					SEM	*p* Value
	WS	PF	PF + WS	PFA	PFA + WS		
HCY (%)	77.16 a	74.70 b	76.27 ab	74.71 b	75.75 ab	0.231	0.0019
CCY (%)	76.99 a	75.36 b	76.39 ab	74.96 b	76.44 ab	0.213	0.0108
Breast (%)	44.69	44.50	44.86	44.34	44.06	0.221	0.8091
Legs (%)	27.68	27.66	27.73	27.73	28.07	0.148	0.9156
Wings (%)	9.23	9.44	9.38	9.57	9.29	0.057	0.4181
Back (%)	17.76	17.63	17.83	17.88	18.10	0.143	0.8982
Boneless breast (%)	36.70	35.69	36.89	36.44	35.59	0.253	0.3540
Boneless legs (%)	19.57	19.46	19.59	19.27	19.65	0.142	0.9341

Average values in the same row followed by different lowercase letters differ significantly (*p* < 0.05). WS: wood shavings; PF: plastic flooring; PF + WS: 50% wood shavings and 50% plastic flooring; PFA: plastic flooring with nanotechnological antimicrobial additive; PFA + WS: 50% wood shavings and 50% plastic flooring with nanotechnological antimicrobial additive. SEM: standard error of the mean.

**Table 6 vetsci-12-00031-t006:** Incidence scores of breast calluses, scratches, bruises, and dermatitis on the carcasses of broiler chickens slaughtered at 42 days and raised on different bedding materials.

Variable	Bedding Material					*p* Value
	WS	PF	PF + WS	PFA	PF + WS	
Breast callus	0.000	0.000	0.000	0.000	0.000	1.0000
Recent scratches	0.555 (0.11)	0.588 (0.11)	0.666 (0.11)	0.611 (0.11)	0.500 (0.11)	0.8879
Old scratches	0.277 (0.10)	0.294 (0.11)	0.222 (0.09)	0.444 (0.11)	0.388 (0.11)	0.6340
Bruises	0.722 (0.10)	0.705 (0.11)	0.722 (0.10)	0.722 (0.10)	0.611 (0.11)	0.9392
Dermatitis	0.888 (0.07)	0.882 (0.07)	0.944 (0.05)	0.944 (0.05)	0.944 (0.05)	0.9125

Values inside parentheses represent the standard error of each estimated mean. WS: wood shavings; PF: plastic flooring; PF + WS: 50% wood shavings and 50% plastic flooring; PFA: plastic flooring with nanotechnological antimicrobial additive; PF + WS: 50% wood shavings and 50% plastic flooring with nanotechnological antimicrobial additive.

**Table 7 vetsci-12-00031-t007:** Dimensions, pH, red content (a*), yellow content (b*), luminosity (L*), water retention capacity (WRC), drip loss (DL), thawing loss (TL), cooking loss (CL), and shear force (SF) of broiler chicken breast fillets raised on different bedding materials at 42 days.

Variable	Bedding Material					SEM	*p* Value
	WS	PF	PF + WS	PFA	PFA + WS		
Length (cm)	17.87 a	17.13 b	17.89 a	17.67 ab	17.77 ab	0.083	0.0282
Width (cm)	9.01 a	8.34 b	9.02 a	8.36 b	9.03 a	0.068	<0.0001
Thickness (mm)	42.15 a	38.63 bc	41.57 ab	37.62 c	41.49 ab	0.427	0.0007
pH	5.97 a	5.91 ab	5.94 ab	5.88 b	5.97 a	0.009	0.0051
a*	3.50 b	4.58 a	3.80 ab	4.19 ab	3.11 b	0.131	0.0028
b*	7.72 a	6.53 ab	7.35 ab	6.25 b	7.74 a	0.173	0.0096
L*	48.63	48.39	48.97	49.03	49.08	0.258	0.9105
WRC (%)	31.18	30.31	30.52	30.49	29.55	0.315	0.6023
DL (%)	4	3.62	3.74	3.71	3.45	0.113	0.6413
TL (%)	5.59	5.1	5.06	5.34	4.84	0.192	0.7819
CL (%)	24.96	26.23	25.91	25.41	25.49	0.293	0.7214
SF (kgf/cm^2^)	1.53	1.68	1.62	1.66	1.66	0.036	0.7239

Average values in the same row followed by different lowercase letters differ significantly (*p* < 0.05). WS: wood shavings; PF: plastic flooring; PF + WS: 50% wood shavings and 50% plastic flooring; PFA: plastic flooring with nanotechnological antimicrobial additive; PFA + WS: 50% wood shavings and 50% plastic flooring with nanotechnological antimicrobial additive. SEM: standard error of the mean.

## Data Availability

Data will be available when requested from the corresponding author.

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
