# Peer review of "Nanotechnological Plastic Flooring: Implications for Broiler Chicken Performance, Health, and Carcass Quality"

_vetsci, 2025, doi:10.3390/vetsci12010031_

Round 1
Reviewer 1 Report
Comments and Suggestions for Authors
I have a few inquiries:
1) Some of the data can be better presented. For example, the first figure showing the pens is not very clear. Pictures taken from the same angle, or a scheme may be more efficient for understanding. Moreover, there are many tables listing a lot of data. It would be more helpful to draw graphs like figure 5 for readers to understand the trend.
2) There are some terms that are not very precise. For example, the authors “inorganic metal ions, such as zinc oxide” at line 100 on page 2.
3) It may be helpful to include the information of the materials for the flooring for future reference and repetition of the experiment. Especially, the authors should explain the composition and structure of PFA in detail. They should also explain why they are interested in such material and how they expect the nanomaterial will work to affect the performance, yield, meat quality, and microbiology since it is the focus of the study as stated in the title. Furthermore, they should explain why the expectation is not met at the end.
4) There is not enough discussion of the cause of certain results. The authors explained that for the PFA, “the effectiveness of antimicrobials depends on several factors”, but the authors only stated one.
5) There are many results listed but not discussed carefully. As authors described, “the weight of the gizzard, which is closely tied to muscular activity and food consumption”. However, very limited explanations are provided, tying the results with the litter materials. Also, there should be reference supporting this. Moreover, “birds raised on plastic flooring were less mobile in terms of access to feed and spent more time sitting” is described. They attributed it to the low temperature, but they didn’t state whether it would be the same case for higher temperatures.
6) As the results indicate, there are not many cases of Eimeria observed. However, there is no explanation of the source. I also wonder whether it would be helpful to introduce controlled amounts of Eimeria to the pens to see the resistance of PFA to the microbes.
Comments on the Quality of English Language
The quality of the language can be improved.
Author Response
Dear Reviewer,
Thank you for your suggestions.
In order to improve the quality of our manuscript, we seek to respond as best as possible to the suggestions made.
The corrections made are marked in the text in red and indicated in the body of this letter.
The manuscript also underwent a new English revision.
Best regards,
Reviewer 1
Dear Reviewer,
Thank you for your suggestions.
In order to improve the quality of our manuscript, we seek to respond as best as possible to the suggestions made.
The corrections made are marked in the text in red and indicated in the body of this letter.
The manuscript also underwent a new English revision.
Best regards,
Reviewer 1
1) Some of the data can be better presented. For example, the first figure showing the pens is not very clear. Pictures taken from the same angle, or a scheme may be more efficient for understanding. Moreover, there are many tables listing a lot of data. It would be more helpful to draw graphs like figure 5 for readers to understand the trend.
R: These are the best photos that we have. We opted for tables for results better than graphics.
2) There are some terms that are not very precise. For example, the authors “inorganic metal ions, such as zinc oxide” at line 100 on page 2.
R: The term was changed in the text.
3) It may be helpful to include the information of the materials for the flooring for future reference and repetition of the experiment. Especially, the authors should explain the composition and structure of PFA in detail. They should also explain why they are interested in such material and how they expect the nanomaterial will work to affect the performance, yield, meat quality, and microbiology since it is the focus of the study as stated in the title. Furthermore, they should explain why the expectation is not met at the end.
R: Complementary information was included in the introduction, description of bedding materials and conclusion sections.
4) There is not enough discussion of the cause of certain results. The authors explained that for the PFA, “the effectiveness of antimicrobials depends on several factors”, but the authors only stated one.
R: Complementary information was included in the discussion.
5) There are many results listed but not discussed carefully. As authors described, “the weight of the gizzard, which is closely tied to muscular activity and food consumption”. However, very limited explanations are provided, tying the results with the litter materials. Also, there should be reference supporting this. Moreover, “birds raised on plastic flooring were less mobile in terms of access to feed and spent more time sitting” is described. They attributed it to the low temperature, but they didn’t state whether it would be the same case for higher temperatures.
R: Complementary information was included in the discussion.
6) As the results indicate, there are not many cases of Eimeria observed. However, there is no explanation of the source. I also wonder whether it would be helpful to introduce controlled amounts of Eimeria to the pens to see the resistance of PFA to the microbes.
R: In the present study we were not able to introduce Eimeria, but we agree that for a future study it would be essential to work on this challenge.

Reviewer 2 Report
Comments and Suggestions for Authors
Hello dear Dr,
The important points related to the manuscript entitled “Nanotechnological plastic flooring: Implications for broiler ...…” are as follows;
Major
· Given that similar studies have been done, state the innovation of this study.
Minor
Introduction
· L 47-88: Please summarize. It seems that it is possible to make the text shorter.
· L 85: Please add reference
Abstract
· L 26-28: For this issue, have the economic aspects also been examined?
Materials and methods
· L 114-117: Is this explanation necessary?
· L 147 "and had antibacterial and antifungal properties": How have you evaluated and confirmed these features?
RESULT
· What is meant by MSE in the tables of this manuscript?
· Is there any information about intestinal histopathology parameters in this experiment?
References
· Please check the references again. The writing format should be done according to the journal guidelines for all references.
Best regards,

Author Response
Dear Reviewer,
Thank you for your suggestions.
In order to improve the quality of our manuscript, we seek to respond as best as possible to the suggestions made.
The corrections made are marked in the text in red and indicated in the body of this letter.
The manuscript also underwent a new English revision.
Best regards,
Reviewer 2
The important points related to the manuscript entitled “Nanotechnological plastic flooring: Implications for broiler ...…” are as follows;
Major
Given that similar studies have been done, state the innovation of this study.
R: Aditional information was included in introduction section stating study innovation.
Minor
Introduction
L 47-88: Please summarize. It seems that it is possible to make the text shorter.
R: this section was reviewed an sumarized
L 85: Please add reference
R: Reference added
Abstract
L 26-28: For this issue, have the economic aspects also been examined?
R: In this study we did not perform an economic evaluation. We are aware of its fundamental importance and it should be included in future studies.
Materials and methods
L 114-117: Is this explanation necessary?
R: We believe that characterizing the study site is important to situate the reader in our region and location within a continental country like Brazil.
L 147 "and had antibacterial and antifungal properties": How have you evaluated and confirmed these features?
R: Sentence reformulated.
RESULT
What is meant by MSE in the tables of this manuscript?
R: It was a mistake. The correct form is SEM: standard error means
Is there any information about intestinal histopathology parameters in this experiment?
R: For this study we did not perform histopathological analysis. We are aware of its fundamental importance and it should be included in future studies.
References
Please check the references again. The writing format should be done according to the journal guidelines for all references.
R: references were reviewed

Reviewer 3 Report
Comments and Suggestions for Authors
Dear Authors,
The article “Nanotechnological plastic flooring: Implications for broiler chicken performance, health, and carcass quality” raises an interesting topic related to the use of nanotechnology in broiler substrates, which may arouse interest in the breeding industry. Thematically, it fits the scope and purpose of the journal.
The structure and layout of the article are correct. The introduction and review of the literature should be expanded to include additional aspects influencing the welfare of birds, such as the location of the facility (the impact of the external climate) and technical solutions, as described, for example, in the article 10.3390/en14248565.
The purpose of the work has been clearly and precisely formulated.
The methodology was described in detail and in a way that allows the repetition of the described experiment. However, please explain why long-term studies, e.g. year-round studies, were not carried out? I believe that the impact of external microclimate conditions may be significant for the obtained research results. Please also explain what criteria the Authors used when selecting specific materials for the litter. I also ask for information on the material solutions of the entire floor (was it thermally insulated?). The research results correspond to the assumptions of the methodology. As in the case of the methodology, please explain how the external microclimate and floor temperature could affect the welfare of the birds?
It is worth expanding the research discussion to include the issue related to the impact of floor material solutions on the emission of greenhouse gases into the atmosphere, referring, for example, to paper 10.1016/j.jobe.2023.107422
The conclusions in the article refer to the stated goal of the research and the results obtained. They are both scientific and practical in nature.
Overall, I rate the substantive and scientific quality of the article as good. After making appropriate corrections and after explaining the aforementioned issues in the review, the article will be eligible for publication.
Author Response
Dear Reviewer,
Thank you for your suggestions.
In order to improve the quality of our manuscript, we seek to respond as best as possible to the suggestions made.
The corrections made are marked in the text in red and indicated in the body of this letter.
The manuscript also underwent a new English revision.
Best regards,
Reviewer 3
Dear Authors,
The article “Nanotechnological plastic flooring: Implications for broiler chicken performance, health, and carcass quality” raises an interesting topic related to the use of nanotechnology in broiler substrates, which may arouse interest in the breeding industry. Thematically, it fits the scope and purpose of the journal.
The structure and layout of the article are correct. The introduction and review of the literature should be expanded to include additional aspects influencing the welfare of birds, such as the location of the facility (the impact of the external climate) and technical solutions, as described, for example, in the article 10.3390/en14248565.
The purpose of the work has been clearly and precisely formulated.
The methodology was described in detail and in a way that allows the repetition of the described experiment. However, please explain why long-term studies, e.g. year-round studies, were not carried out?
R: Since this is a bedding material under development, we decided to initially carry out only one production cycle and evaluate the possible responses. This way, we could make adjustments to the material, if necessary. In this way, we believe that the development of the product could be more accurate. We emphasize that long-term studies are in our future plans.
It is worth expanding the research discussion to include the issue related to the impact of floor material solutions on the emission of greenhouse gases into the atmosphere, referring, for example, to paper 10.1016/j.jobe.2023.107422
R: Due to the importance of the topic, we carried out a more specific study on this subject using this same material and it has been published (10.3390/su151712896), so this topic was not part of this manuscript.
The conclusions in the article refer to the stated goal of the research and the results obtained. They are both scientific and practical in nature.
Overall, I rate the substantive and scientific quality of the article as good. After making appropriate corrections and after explaining the aforementioned issues in the review, the article will be eligible for publication.
R: Thank you for your suggestions. They were much valuable to improve our manuscript.
Round 2
Reviewer 1 Report
Comments and Suggestions for Authors
Thank you for adding the details and explanations for my questions. The overall explanation is reasonable, but it is crucial to be clearer in your writing since a lot of information is presented, but the conclusions cannot be drawn from the current results. Especially, please revise your introduction and conclusion part for audience to better follow your logics. Also, there are a few things that I noticed.
1) "Birds raised on conventional litter are more exposed to environments with inferior sanitary conditions" at line 60 is confusing. Does conventional litter introduces the microbes or it helps the spread?
2) Is it "food poisoning risk" at line 65?
3) First sentence in line 73 is redundant. Delete, or move to the beginning.
4) Equations of Figure 5 can be better formatted.
5) Descirption fo Figure 6 can be better formatted (line 436-440).
6) Since zinc oxide antimicrobial additive is the crucial part of your study, especially it's in the title, can you provide more detailed explanations of it's influence on your results and how it can be impactful to the field?
Author Response
Dear Reviewer,
Thank you for your suggestions.
In order to improve the quality of our manuscript, we seek to respond as best as possible to the suggestions made.
The corrections made are marked in the text in red and indicated in the body of this letter.
Best regards,

Reviewer 3 Report
Comments and Suggestions for Authors
The authors responded to the comments in the review and made corrections to the article. I recommend publishing the article in its current form.
Author Response
Dear Reviewer,
Thank you for your suggestions.
In order to improve the quality of our manuscript, we seek to respond as best as possible to the suggestions made.
Best regards,